# Unraveling the significance of decorin in endometriosis development through single cell sequencing and experimental approaches

Guansheng Chen⊙, Wenjing Li⊙, Lingyu Liu, Yongjun Wang⊙*

Department of Gynecology and Obstetrics, Beijing Jishuitan Hospital, Capital Medical University, Beijing, China

⊙ These authors contributed equally to this work.
* wangyongjun@jst-hosp.com.cn

## Abstract

### Background

A link exists between decorin (DCN) and endometriosis, nevertheless, the role of DCN in this condition remains unclear. This study aims to bioinformatically characterize DCN expression, diagnostic value, and related signaling in endometriosis using single-cell and bulk transcriptomic data, with experimental validation of expression level.

### Methods

Single-cell RNA sequencing (scRNA-seq) was analyzed using Scissor and ROGUE algorithms to identify key cell types associated with endometriosis. CIBERSORTx was used to construct a signature matrix for deconvolution of bulk RNA-seq data and estimation of immune and stromal subpopulation proportions. DCN expression was explored across datasets and validated by reverse transcription-quantitative polymerase chain reaction (RT-qPCR) in endometrial and endometriotic stromal cells. Its diagnostic value was assessed by receiver operating characteristic (ROC) curve analysis, and associated pathways were predicted by gene set enrichment analysis (GSEA). Potential DCN-targeted drugs were predicted via bioinformatic databases.

### Results

Stromal cells were identified as the key cell type with high heterogeneity and dominant DCN expression. Ten stromal subtypes were delineated, with Activated/Myofibroblast-like Stromal Cells (AMSC), Fibrosis-Effector Stromal Cells (FESC), Stemness-Remodeling Stromal Cells (SRSC), and dStromal showing significantly altered proportions in endometriosis. DCN was significantly upregulated in endometriosis at both transcriptomic and cellular levels, with area under the curve (AUC) > 0.8 in two independent datasets, supporting its diagnostic potential. GSEA indicated DCN

**Data availability statement:** All relevant data are within the manuscript and its Supporting information files.

**Funding:** Beijing Jishuitan Hospital Youcai Plan (JSTYC202402).

**Competing interests:** The authors have declared that no competing interests exist.

correlates with complement/coagulation cascades, TGF-β signaling, and other pathways linked to tissue remodeling and inflammation. Seven candidate drugs targeting DCN were predicted as hypothesis-generating leads.

## Conclusion

This study provides bioinformatic and correlative evidence that DCN is abnormally upregulated in endometriotic stromal cells and exhibits favorable diagnostic value. DCN is associated with pathways relevant to endometriosis pathogenesis and represents a promising biomarker and therapeutic candidate pending future functional and mechanistic validation.

## 1. Introduction

Endometriosis affects approximately 6–10% of reproductive-aged women and is associated with chronic pelvic pain, dysmenorrhea, infertility, and reduced quality of life [1]. Because it not only causes physical discomfort but also places a considerable psychological and emotional strain on adults and adolescents, this disorder drastically lowers the quality of life for women. Globally, diagnosing endometriosis is still difficult, especially for women who initially exhibit symptoms while they are young. According to the most recent data, it takes about eight years to identify this illness [2]. The etiology of endometriosis remains incompletely understood, though retrograde menstruation, immune dysfunction, hormonal imbalances, and genetic predisposition are implicated in its pathogenesis [3]. Current treatment strategies, including hormonal therapy and surgery, often provide only symptomatic relief and are associated with high recurrence rates [4]. Therefore, identification and characterization of novel candidate biomarkers with diagnostic and potential therapeutic relevance for endometriosis remains a critical research priority.

Decorin (DCN), a small leucine-rich proteoglycan, is encoded by the DCN gene located on chromosome 12q21.33 [5]. It is a key component of the extracellular matrix (ECM) and regulates collagen fibrillogenesis, cellular growth, and inflammation [6]. In the corpus luteum, follicular fluid, and theca cells of adult rhesus macaques and human ovaries, DCN expression was detected by immunohistochemical examination [7]. Emerging evidence suggests that DCN plays multifaceted roles in various pathological processes, including fibrosis, cancer, and immune regulation [8,9]. For instance, DCN can inhibit tumor growth and metastasis by modulating signaling pathways such as TGF-β and EGFR [10]. However, the expression profile, clinical relevance and molecular correlates of DCN in endometriosis remain poorly defined. While a preliminary association between DCN and endometriosis has been proposed [11], systematic characterization of DCN's expression pattern, diagnostic potential and associated signaling pathways in endometriosis development and progression is largely lacking.

In this study, we employed a combined approach of bioinformatics analysis and experiments to investigate the expression characteristics, diagnostic value, and

related signaling pathways of DCN in endometriosis, and conducted experimental verification of its expression levels. We first identified endometriosis-associated key cell types from scRNA-seq data and performed secondary clustering to delineate stromal cell heterogeneity. A custom signature matrix was constructed and applied to deconvolute bulk transcriptomic data, enabling comparative analysis of the cellular microenvironment between endometriosis and control samples. We further characterized the expression pattern of DCN and validated its differential expression via reverse transcription-quantitative polymerase chain reaction (RT-qPCR) in endometrial and endometriotic stromal cells. Additionally, we explored the diagnostic potential of DCN and its associated molecular pathways in endometriosis. This work provides a comprehensive characterization of DCN in endometriosis, laying the groundwork for future functional studies to elucidate its mechanistic role.

## 2. Materials and methods

### 2.1 Data source

Endometriosis-related transcriptome (GSE135485 and GSE23339), single-cell RNA sequencing (scRNA-seq) datasets (GSE179640) and the independently verified dataset (GSE7307) were all downloaded from the Gene Expression Omnibus (GEO, https://www.ncbi.nlm.nih.gov/geo/) database. GSE135485 dataset, including 54 endometriosis and 4 control endometrium samples, was sequenced by GPL21290 platform. Furthermore, GSE23339 dataset (GPL6102), comprised of 10 endometriosis and 9 control endometrium samples. GSE179640 dataset included 28 endometriosis and 3 control samples, sequencing by GPL24676. Among 28 endometriosis samples, 9 eutopic endometrium, 8 ectopic peritoneal, 7 ectopic peritoneal adjacent, and 4 ectopic ovary of endometriosis patients were included. GSE7307 dataset, including 18 endometriosis and 23 control endometrium samples, was sequenced by GPL570 platform.

### 2.2 ScRNA-seq analysis

The scRNA-seq data were processed for quality control using "Seurat" package (version 5.1.0) [12] with the following exclusion criteria: (1) genes covered in less than 200 cells, (2) cells with genes ≥ 8,000, (3) genes with count ≥ 50,000, and (4) cells with mitochondrial genes ≥ 20%. Subsequently, vst method was used to extract top 2,000 highly variable genes. Single-cell data were scaled using ScaleData function, and then used to identify statistically significant principle components (PCs) by JackStrawPlot function. Principle component analysis (PCA) was employed to calculate the significance among highly variable genes and PCs using RunPCA function, and top 30 PCs were utilized for further analysis. FindNeighbors and FindClusters functions in "Seurat" package were used to identify cell clusters at resolution = 0.1, which were visualized by uniform manifold approximation and projection (UMAP) method. According to the marker genes in published literature [13], cell types were annotated. Furthermore, "DoubletFinder" package (version 2.0.4) [14] was utilized to identify and exclude doublets to diminish the influence of them for analysis. Finally, the proportion of cell types in endometriosis and control samples was compared.

### 2.3 Identification and analysis for key cells

The "Scissor" package (version 2.0.0) [14] was used to determine endometriosis-related cells by integrating scRNA-seq and bulk RNA data. The parameters for Scissor function were "alpha" = 0.0001 and "family" = "binomial". Then, Scissor+ (positively correlated with endometriosis) and Scissor- (negatively correlated with endometriosis) cells were identified. Additionally, "ROGUE" package (version 1.0) [15] was used to calculate the heterogeneity purity score. The expression of DCN was investigated in cell types between endometriosis and control samples. By combining the results of Scissor, heterogeneity, and expression analyses, the key cell was selected for secondary clustering. FindNeighbors and FindClusters functions were used for unsupervised clustering analysis on the key cell with resolution = 0.2, and visualized by UMAP method. Based on marker genes and functions, cell subtypes were annotated.

 

## 2.4 Pseudotime analysis and cell communication

To draft the differentiation within endometriosis progression, pseudotime analysis of key cell was performed using "monocle" package (version 2.26.0) [16]. The expression of DCN during differentiation was also explored. Additionally, cell communication analysis was conducted to investigate the interactions among cell types by "cellchat" package (version 1.6.1) [17].

## 2.5 Construction and validation of signature matrix

All cell types were extracted and sampled at 1,000, divided into training and test cohorts within a 7:3 ratio. The annotated scRNA-seq data was uploaded to CIBERSORTx to generate a signature matrix using "Create Signature Matrix" function with the parameter as default in the training cohort. Subsequently, the performance of signature matrix was verified in the test cohort. Based on signature matrix, the "Impute Cell Fractions" function in CIBERSORTx was used to evaluate the proportion of cells. Pearson analysis was conducted between estimated and actual percentage. CIBERSORTx and signature matrix were subsequently used to perform deconvolution in both GSE135485 and GSE23339 datasets, and the difference of all cell types was compared between endometriosis and control samples using Wilcoxon test ($P < 0.05$). In the GSE135485 dataset, the correlation of DCN with all cell types was also explored.

## 2.6 The diagnostic value and functional analysis of DCN in endometriosis

To determine the crucial role of DCN in endometriosis, the expression of this gene was compared between control and disease groups, as well as diagnostic value of this gene was evaluated by receiver operating characteristic (ROC) curves across two datasets and one independent dataset (GSE7307). Furthermore, the potential function of DCN was explored by gene set enrichment analysis (GSEA) utilizing "clusterProfiler" package (version 4.6.2) [18]. Spearman analysis was performed between DCN and all genes, and the correlation coefficient was the ranked threshold. Background gene set was obtained from GSEA website (http://www.gsea-MSigdb.org/gsea/msigdb). Screening criteria for GSEA were |NES| > 1 and $P < 0.05$.

## 2.7 Regulatory relationship and associated drugs targeting DCN

To investigate the regulatory mechanisms of DCN, corresponding miRNAs and lncRNAs were predicted by miRDB (http://mirdb.org) and starbase databases (https://rnasysu.com/encori/), respectively. The screening thresholds for miRNAs and lncRNAs were respectively score > 60 and clipExpNum > 10, respectively. Based on these results, the DCN-miRNA-lncRNA regulatory network was constructed. Potential drugs for endometriosis were predicted by targeting DCN using DGIDB (https://coremine.com/medical/).

## 2.8 Cell culture

The human Endometrial stromal cells hESC and Endometriosis stromal cells hEM15A were provided by Dr. Y. Shimada (Kyoto University, Japan). Prior to the study, all cell lines were identified by short tandem repeat DNA fingerprinting at Peking Union Medical College (Beijing, China). These cells were cultured in DMEM/F12 (Gibco, USA) medium (HyClone, Logan, UT, USA) with 1% penicillin/streptomycin and 10% fetal bovine serum at a temperature of 37°C, with 5% $CO_2$.

## 2.9 Reverse transcription-quantitative polymerase chain reaction (RT-qPCR)

Total RNA was isolated from hESC and hEM15A using Trizol reagent (Invitrogen) and reverse-transcribed into cDNA. The RT-qPCR was conducted according to the manufacturer's instructions using Hiscript II QRT Supermix for qPCR kits (Vazyme, China). Biological and technical triplicates were conducted to validate the reliability of results. The mRNA

levels were standardized using GAPDH. The relative mRNA levels were calculated and normalized by $2^{-\Delta\Delta Ct}$ method. The sequences of primers were contained in S1 Table.

## 2.10 Statistical analysis

All statistical analyses for bioinformatics were conducted employing R software (version 4.1.2). The experimental data were expressed as mean ± standard deviation (SD). Statistical analyses were conducted using unpaired two-tailed Student's t-tests or one-way ANOVA with least significant difference tests. A $P$ value less than 0.05 was considered statistically significant.

## 3. Results

### 3.1 A total of 5 cell types were annotated in endometriosis

After quality control and data filtering, 107,331 cells (13,358 control and 93,973 endometriosis) were obtained. Following the selection of the top 2,000 highly variable genes for dimensionality reduction via PCA (S1A Fig), the top 30 PCs were selected for subsequent analysis (S1B Fig). A total of 19 cell clusters were annotated (Fig 1A), and 5 cell types were yielded. Among these cells, 8,050 doublets (7.5%) were removed, leaving 99,281 cells with clearer cell boundaries (Fig 1B

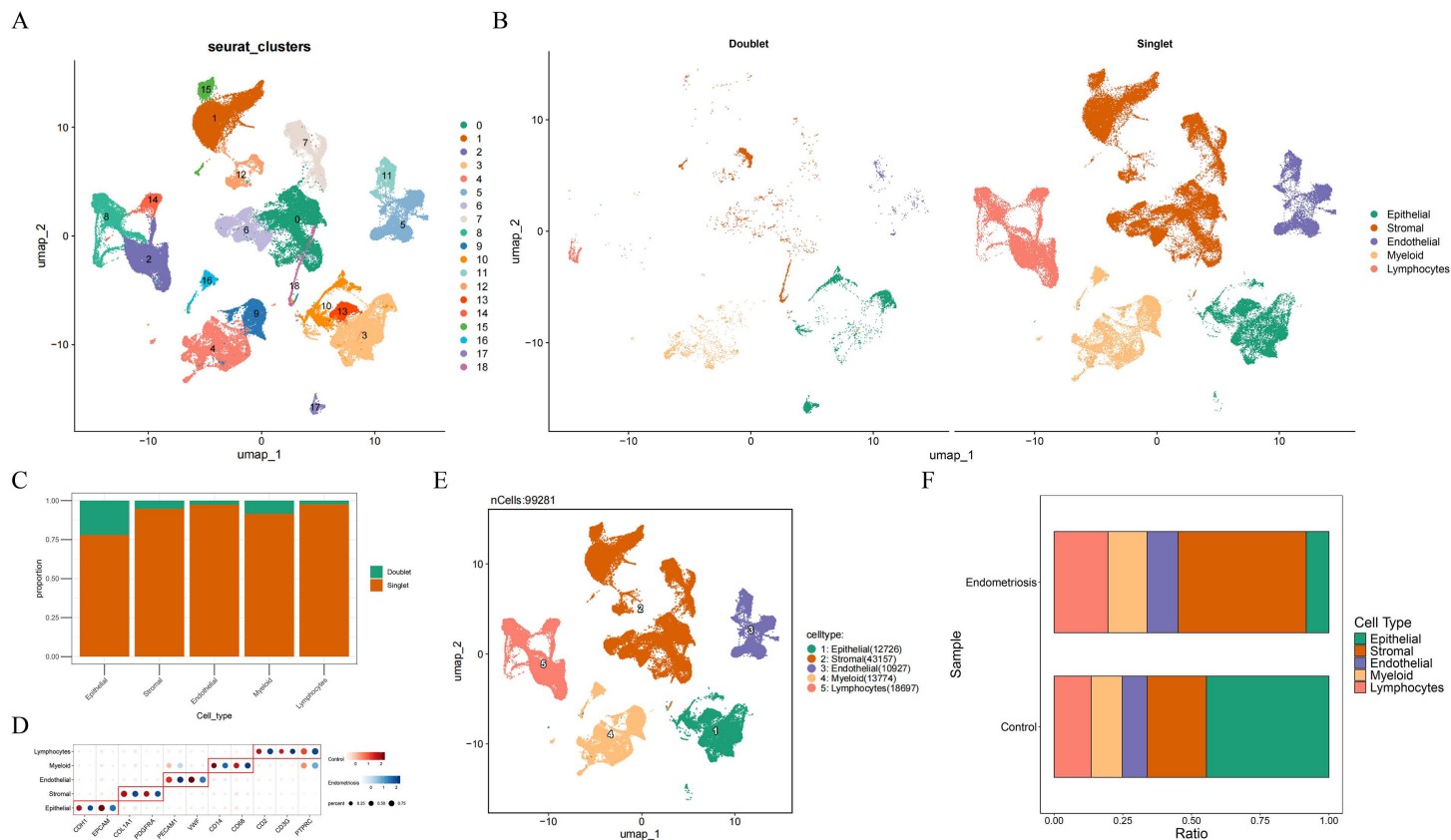

**Fig 1. A total of 5 cell types were annotated in endometriosis. (A)** The identification of cell clusters in single-cell data. **(B)** The visualization of singlet and doublets. **(C)** The proportion of doublets. **(D)** The marker genes for cell types. **(E)** The annotation of cell types. **(F)** The proportion of cell types in control and endometriosis samples.

and 1C). These 5 cell types were epithelial, stromal, endothelial, myeloid, and lymphocyte cells (Fig 1D and 1E). Proportion analysis revealed that stromal cell had the highest percentage in endometriosis samples (Fig 1F).

### 3.2 Stromal cells were identified as key cells

To identify endometriosis-related cells, Scissor analysis was conducted, resulting in the endometriosis positively correlated cells (Scissor+: stromal, endothelial, myeloid, and lymphocytes cells) and negatively correlated cells (Scissor-: stromal and epithelial cells) (Fig 2A and 2B). These results showed that stromal cells exhibited heterogeneity in endometriosis samples. Furthermore, stromal cells had the low ROGUE score, further indicating the higher heterogeneity (Fig 2C). Additionally, the expression of DCN was found to be the highest in stromal cells (Fig 2D). By combining the high heterogeneity, the highest expression of DCN, and stromal cells were found to influence the development of endometriosis [13,19,20], stromal cells were selected as the key cell for subsequent analysis. Secondary clustering analysis identified 14 cell subclusters for stromal cells (Fig 2E), and marker genes for each subtype were listed in S2 Table. According to marker genes, cluster 4/12 was annotated as eStromal [20], cluster 1/2/3/8 were annotated as dStromal [20], cluster 6 was annotated as vascular smooth muscle (VSMC) [21], and cluster 7 was annotated as pericytes [19]. The above subpopulations are consistent with the classic endometrial stromal subtypes reported in the literature in terms of signature gene expression and biological functions. For remaining cell subclusters, FindMarkers function was used to select differentially expressed genes (DEGs) with $|\log_2 FC| > 0.5$ and adj.$P < 0.05$. GO and KEGG enrichment analyses were conducted to assist in functional annotation. Relied on functions, cluster 0 was annotated as Pre-Synthetic Stromal Cell (HSSC), cluster 5 was annotated as Activated/Myofibroblast-like Stromal Cells (AMSC), cluster 9 was annotated as WNT-EMT Activated Stromal Cells (WEASC), cluster 10 was annotated as Fibrosis-Effector Stromal Cells (FESC), cluster 11 was annotated as Stemness-Remodeling Stromal Cells (SRSC), cluster 13 was annotated as Metabolic-Endocrine Dual-Function Stromal Cells (MEDSC) (Fig 2F and 2G). Totally, 10 cell subtypes were annotated within stromal cells.

### 3.3 Expression of DCN in the early and middle stage of stromal differentiation

Proportion analysis revealed that dStromal and HSSC had higher proportions in endometriosis (Fig 3A). Pseudotime analysis of stromal cells showed that there were 5 stages existed during differentiation, with all stages were existing in endometriosis samples (Fig 3B). Furthermore, DCN was expressed in the early and middle differentiation stages (Fig 3C). Cell communication analysis was conducted on cell types, and it was evident that the number and strength of interactions were increased significantly in endometriosis (Fig 3D and 3E). This indicated the frequent interactions across cells in disease.

### 3.4 Signature matrix well predicted the proportion of cellular composition

To validate the performance of the signature matrix, deconvolution was processed using CIBERSORTx (Fig 4A and S3 Table). A significant correlation was existed in estimated and actual cell populations (cor=0.59, $P < 0.05$), which indicated that custom signature matrix well predicted cell proportion (Fig 4B). For both transcriptome datasets, CIBERSORTx and signature matrix were used to conduct deconvolution. Cellular composition proportion was estimated in both control and endometriosis samples (Fig 4C and 4D). The differential analysis showed that AMSC, dStromal, FESC, and SRSC had the significant difference between control and endometriosis samples for both sets (Fig 4E and 4F). In the GSE135485 dataset, the correlation of cellular composition with DCN was explored, and the results showed that DCN positively correlated with AMSC, SRSC, FESC, HSSC, and endothelial cells, while negatively associated with dStromal, eStromal, and epithelial cells (Fig 4G).

### 3.5 Functional and regulatory roles of DCN in endometriosis

To deeper understand the role of DCN in the development of endometriosis, specific analysis was conducted on DCN. The expression of DCN between control and endometriosis was compared, with the higher expression in disease group

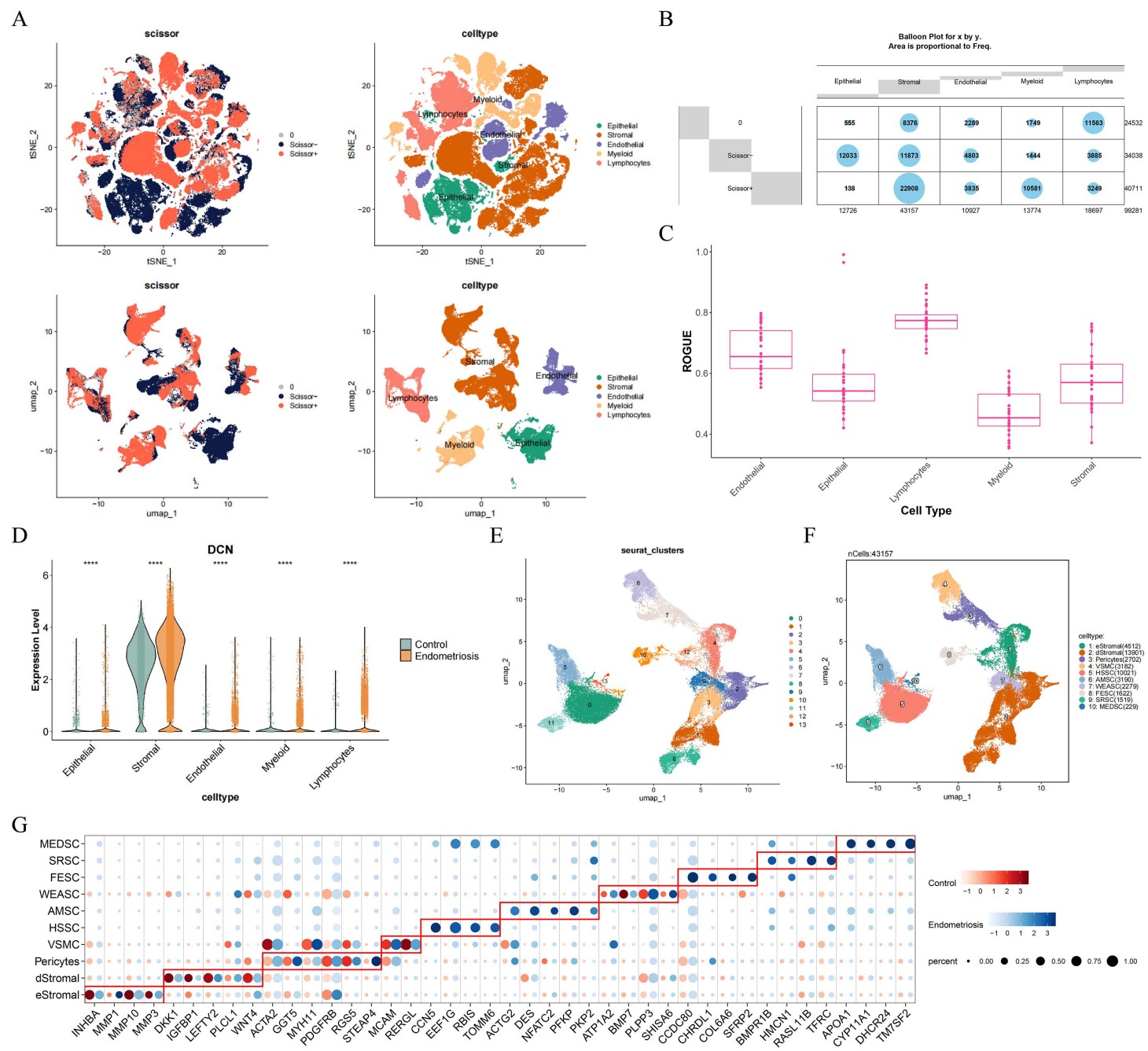

**Fig 2. Stromal cell was identified as key cell. (A)** The distribution of cells positively or negatively correlated with endometriosis. **(B)** The classification of cell types. **(C)** The heterogeneity analysis of cell types. **(D)** The expression of DCN in cell types. **(E)** The identification of cell subclusters. **(F)** The annotation of cell subtypes. **(G)** The marker genes for cell subtypes. ****$P < 0.0001$.

within both datasets (Fig 5A). Further RT-qPCR validated the higher expression of DCN in hEM15A cells (Fig 5B). ROC curve showed that area under the curve (AUC) value of DCN for both datasets was over 0.8 (GSE135485: 0.884, GSE23339: 0.867) (Fig 5C), implying that DCN could serve as a promising predictive indicator for endometriosis. In

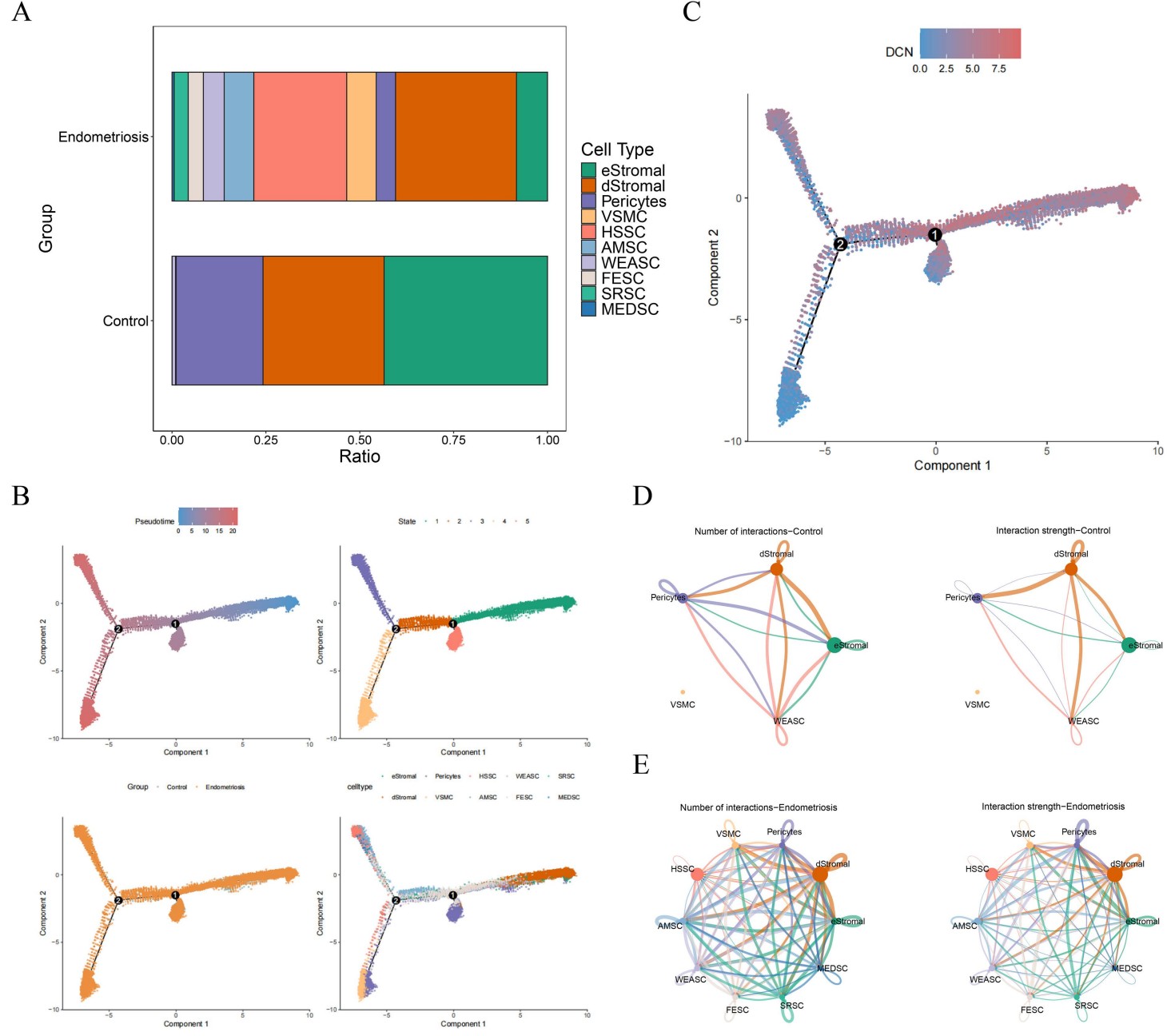

**Fig 3. Pseudotime analysis and cell communication of cell types. (A)** The proportion of cell subtypes in control and endometriosis groups. **(B)** The differentiation of stromal cells. **(C)** The expression of DCN during stromal cell differention. **(D-E)** Communication of cell subtypes in control (D) and endometriosis samples (E).

addition, in the independent dataset GSE7307, the AUC value of DCN also exceeded 0.8 (0.814) (S2 Fig), demonstrating the reliability of the results. GSEA of DCN revealed the enrichment of 34 pathways, and top 6 pathways of significance included complement and coagulation cascades, DNA replication, proteasome, ribosome, and TGF-β signaling pathway (Fig 5D). These pathways played a crucial role in the damage and repair of diseases. Based on DCN, 626 miRNAs and

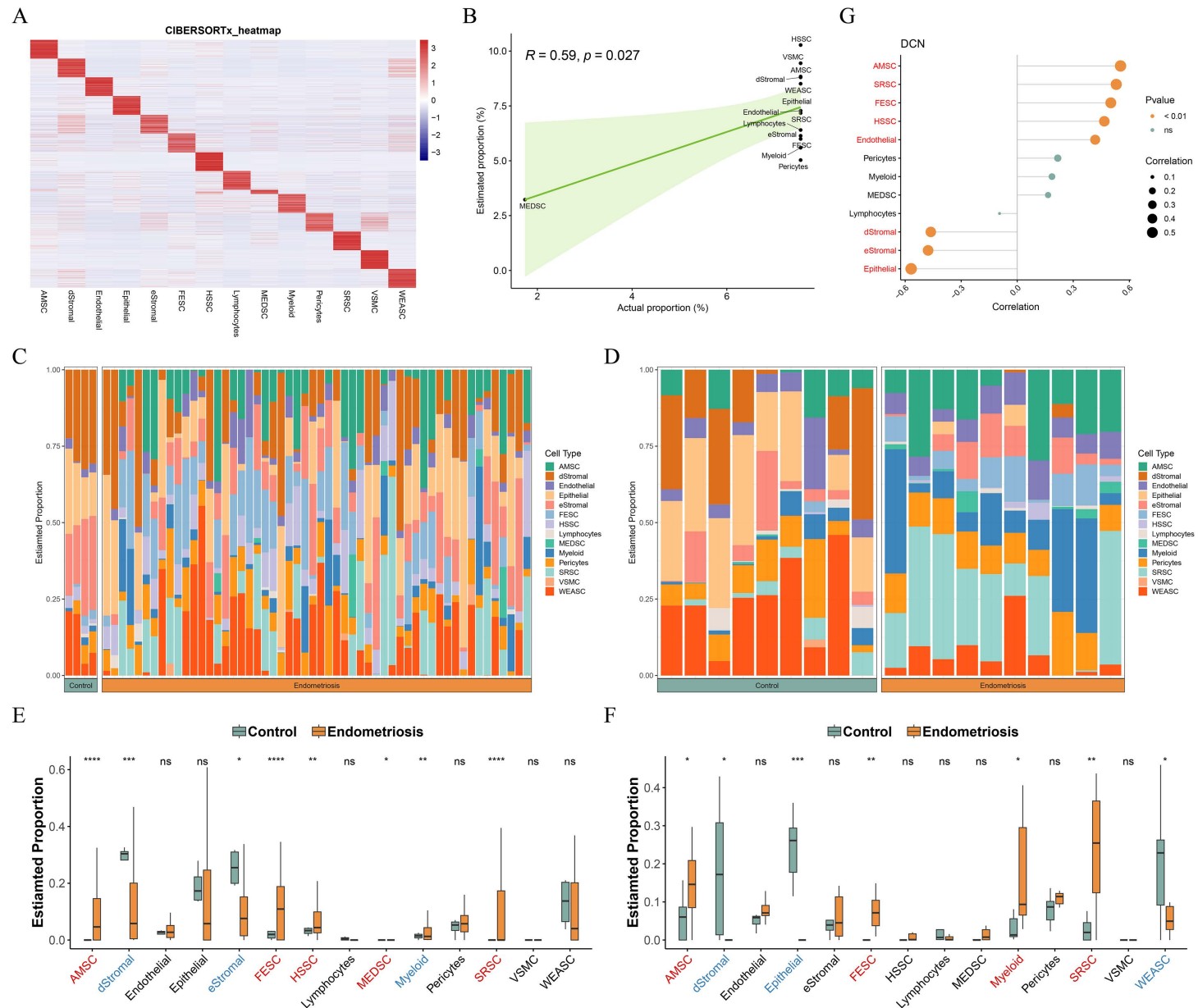

**Fig 4. Signature matrix well predicted the proportion of cellular composition. (A)** The heatmap of cellular composition proportion. **(B)** Correlation of estimated and actual cellular composition proportion. **(C-D)** Cellular composition diagram in both GSE135485 (C) and GSE23339 datatsets (D). **(E-F)** The difference of cellular composition between control and endometriosis samples in both GSE135485 (E) and GSE23339 datatsets (F). **(G)** Correlation between DCN and cellular composition. ns, no significance, *P < 0.05, **P < 0.01, ***P < 0.001, ****P < 0.0001.

75 miRNA-lncRNA interaction pairs were predicted, resulting in the construction of a mRNA-miRNA-lncRNA regulatory network (Fig 5E). In this network, DCN-hsa-miR-103a-3p-AC093297.2, DCN-hsa-miR-642a-3p-NORAD interactions were found. Additionally, seven drugs, including marimastat, CM-352, recombinant heat shock proteins, recombinant interferon, ascorbic acid, sirolimus, and the MMP13 tracer, were predicted to target DCN (Fig 5F). These drugs may serve as candidate molecules worthy of further investigation for endometriosis treatment.

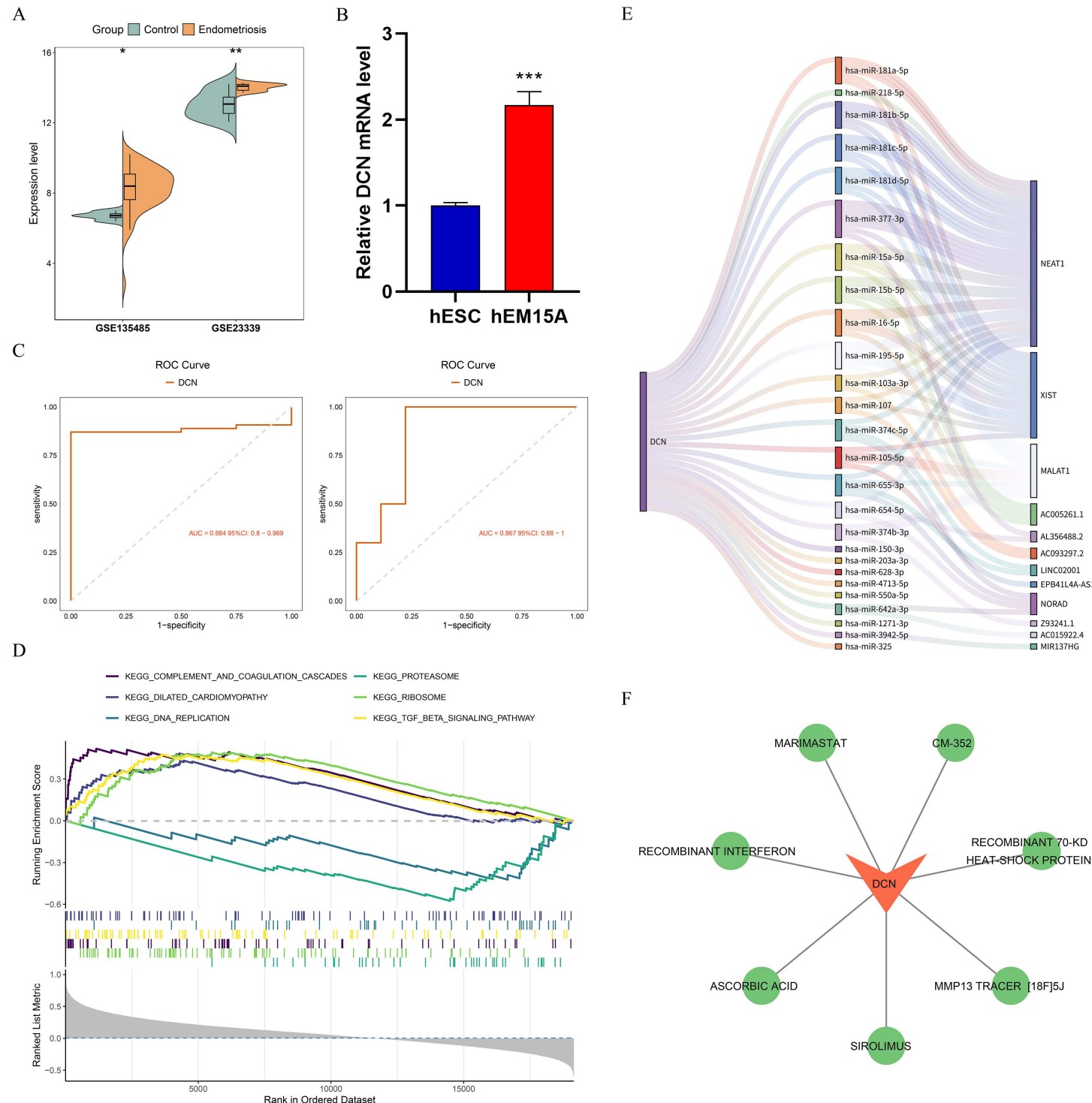

**Fig 5. Functional and regulatory roles of DCN in endometriosis. (A)** The expression of DCN between control and endometriosis samples. **(B)** The expression of DCN between control and endometriosis cells through reverse transcription-quantitative polymerase chain reaction (RT-qPCR). **(C)** Receiver operating characteristic (ROC) curve for DCN. **(D)** Gene set enrichment analysis (GSEA) of DCN. **(E)** Regulatory network of DCN. **(F)** Potential drugs predicted by DCN. *$P < 0.05$, **$P < 0.01$, ***$P < 0.001$, AUC, area under the curve.

## 4. Discussion

Endometriosis is a complicated chronic illness that causes a continuous inflammatory response because it involves endometrium-like tissue outside the uterus [22]. Angiogenesis serves as a key driver in the occurrence and progression of endometriosis, and its role in disease progression is analogous to its pivotal status in tumor-like proliferative disorders. As molecules closely associated with angiogenesis, DCN and vascular endothelial growth factor (VEGF) warrant in-depth investigation regarding their correlation with endometriosis. Aydin GA et al. conducted a descriptive study to evaluate the immunohistochemical (IHC) staining characteristics of DCN and VEGF in ovarian and endometrial tissues between patients with endometriosis and those without the disease. The results demonstrated significant differences in the staining intensity and distribution patterns of DCN and VEGF in endometrial and ovarian tissues between patients with endometriosis and subjects undergoing surgery for other benign gynecological diseases. Moreover, a significant negative correlation was observed between the expression patterns of DCN and VEGF in the tested tissues, suggesting that DCN holds promise as a potential molecular target for therapeutic intervention in endometriosis [23]. Nevertheless, existing studies on the correlation between DCN and endometriosis are still scarce, and its specific mechanism of action has not yet been clarified. Therefore, this study integrated single-cell transcriptomics with functional experiments to further elucidate the role of DCN in endometriosis. We identified stromal cells as a key cellular component in the pathogenesis of endometriosis and further delineated 10 functionally distinct stromal subtypes. Notably, we demonstrated that DCN had significantly differential expression trends between control and endometriotic stromal cells.

Our scRNA-seq analysis reaffirmed the central role of stromal cells in endometriosis, consistent with recent literature highlighting their functional heterogeneity and contribution to disease pathology [24]. Stromal cells represent the principal element of endometriotic lesions, exhibiting hormone-related gene expression [25,26]. Epigenetic anomalies in stromal cells of endometriotic lesions display partial characteristics of ovarian granulosa cells (estradiol biosynthesis) and macrophages (cytokine production) [27]. Estradiol (E2), largely via its receptor ERβ, functions as a pivotal regulator of essential pathogenic processes in endometriosis, promoting lesion viability and eliciting pain-associated inflammation [28]. Moreover, endometriotic lesions demonstrate resistance to the effects of progesterone. As a result, this tissue is devoid of progesterone receptors, resulting in differentiation abnormalities in both the endometrium and endometriotic lesions [28]. Prior research demonstrates that the activation of the AKT pathway, subsequent to progesterone signaling in stromal cells, precipitates rapid decidualization of endometriotic lesions [29,30]. Simultaneously, aberrant activation of AKT diminishes progesterone receptor protein levels in stromal cells, hence facilitating progesterone resistance in endometriosis [31]. The progesterone resistance observed in endometriosis lesions exacerbates lesion proliferation, persistent inflammation, aberrant gene expression, and endometrial rejection [32]. These findings collectively illustrate the essential function of stromal cells in endometriosis. Subsequently, 10 stromal subtypes, including AMSC, FESC, and SRSC, which were significantly elevated in endometriosis were identified, suggesting a complex landscape of stromal activation and differentiation. Particularly, the expansion of FESC and AMSC subpopulations aligns with emerging evidence that myofibroblast activation and fibrotic processes are hallmarks of advanced endometriosis [33]. The pseudotime trajectory further indicated dynamic differentiation of stromal cells within endometriotic lesions, with DCN highly expressed in early and middle stages, suggesting its potential role in driving stromal activation.

GSEA revealed that DCN was associated with several pathways critically involved in endometriosis, most notably the TGF-β signaling pathway and complement and coagulation cascades. TGF-β signaling is a well-established driver of fibrosis, immune evasion, and cellular proliferation in endometriosis [22,34]. Prior research has shown that the β-catenin pathway contributes to TGF-β1-mediated fibrosis in ovarian endometriosis [35], and that TGF-β1 promotes the TGF-β/Smad signaling pathway, causing fibrosis in ectopic endometriosis lesions [36]. All of these results highlight how important TGF-β signaling is in endometriosis. Moreover, the association between DCN and TGF-β pathway activation is particularly intriguing. Under specific circumstances, DCN, an extracellular matrix component, can function as a natural inhibitor of TGF-β [37] by binding to TGF-β1 and partially neutralizing its action [38], thereby reducing tissue fibrosis. Mechanistically,

DCN suppresses the TGF-β1 pathway by preventing TGF-β1-induced SMAD2/3 phosphorylation, and reduces the expression of fibronectin, collagen I, collagen III, and α-SMA. Additionally, enrichment in complement and coagulation cascades underscores the involvement of inflammatory processes, which are characteristic of the endometriotic microenvironment [39]. The complement cascade is essential in inflammatory and immunological responses [40]. In women, the activation of the C3a component of the complement and coagulation cascades during early pregnancy is significantly correlated with a heightened risk of preterm premature rupture of membranes [41]. Golinska M et al. identified the complement system and platelet coagulation as the two predominant routes in endometriosis, both processes being crucial for the natural growth and shedding cycle of the endometrium [42]. The complement system, a facilitator of tissue growth and regeneration, has been associated with the onset of autoimmune disorders and tumor promotion [43,44]. Thus, its dysregulation may enable immune surveillance evasion and promote lesion implantation. The importance of this in endometriosis has been postulated since the 1980s, with increased levels of C1, C3, and C5 identified in the blood and peritoneal fluid of affected women [45,46]. Numerous complement proteins are found in the epithelial cells of endometrial lesions and ovarian cancer tumors, with their local synthesis and deposition linked to the advancement of multiple cancer types [42,47]. These data further substantiate the critical significance of these two routes in endometriosis.

This study not only deepens our understanding of stromal cell heterogeneity in endometriosis, but also provides preliminary molecular evidence supporting DCN as a promising therapeutic target. Nevertheless, the present work was mainly focused on the expression and distribution patterns of DCN in stromal cells, whereas its biological functions in other cell types within the endometriotic microenvironment remain to be fully elucidated. In addition, the specific molecular mechanisms by which DCN participates in the progression of endometriosis were not directly investigated in the current analysis. One of the major limitations of this study is the absence of an independent clinical validation cohort. Therefore, further validation using large-sample and multicenter clinical cohorts is warranted before our findings can be translated into clinical applications. Notably, given its secretory nature, DCN also holds potential as a noninvasive biomarker, which could be detected in serum, peritoneal fluid, or menstrual blood. Exploring the expression and diagnostic value of DCN in these easily accessible bodily fluids would provide a more feasible strategy for future translational research. In future studies, we intend to perform in vitro functional experiments to investigate the effects of DCN on the migration, invasion, and extracellular matrix remodeling of endometrial cells, evaluate the expression of fibrosis-related markers and the activity of the transforming growth factor-β signaling pathway, and carry out in vivo verification assays, with the goal of clarifying the exact role and regulatory mechanism of DCN in endometriotic progression. Despite the above limitations, our results offer novel insights into the molecular pathogenesis of endometriosis and point to potential directions for subsequent mechanistic investigations, biomarker development, and therapeutic exploration.

## 5. Conclusion

In conclusion, our study identified stromal cells as key participants in the pathogenesis of endometriosis. Based on bioinformatic analyses, we observed that DCN was expressed during the early and middle stages of stromal cell differentiation. Furthermore, we applied CIBERSORTx and deconvolution algorithms to construct a cell proportion signature matrix for evaluating cellular composition in transcriptomic datasets. DCN expression was verified at both the transcriptomic and single-cell levels, and ROC curve analysis supported its potential value as a candidate diagnostic biomarker for endometriosis. Collectively, these findings provide novel bioinformatic evidence supporting the potential involvement of DCN in endometriosis, while its precise biological function and regulatory mechanism in disease progression remain to be further elucidated by experimental validation.

## Supporting information

**S1 Fig. Single-cell RNA sequencing (scRNA-seq) analysis.** (A) Top 2,000 highly variable genes. (B) Top 30 principle components (PCs).
(TIF)

**S2 Fig. The receiver operating characteristic (ROC) curve of the DCN gene in the database GSE7307.**
(TIF)

**S1 Table. The sequences for primers.**
(DOCX)

**S2 Table. The complete marker gene list for each cluster.**
(XLSX)

**S3 Table. The signature matrix constructed by CIBERSORTx.**
(XLSX)

## Acknowledgments

We would like to sincerely thank the authors for their scientific contribution.

## Author contributions

**Conceptualization:** Yongjun Wang, Guansheng Chen.

**Data curation:** Yongjun Wang, Guansheng Chen, Wenjing Li.

**Formal analysis:** Wenjing Li, Lingyu Liu.

**Funding acquisition:** Wenjing Li.

**Investigation:** Guansheng Chen.

**Methodology:** Guansheng Chen, Wenjing Li.

**Project administration:** Yongjun Wang.

**Resources:** Yongjun Wang.

**Software:** Lingyu Liu.

**Validation:** Lingyu Liu.

**Writing – original draft:** Guansheng Chen, Wenjing Li.

**Writing – review & editing:** Yongjun Wang, Guansheng Chen.

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
