## [Decision Letter · Decision Letter 0]

20 Feb 2026

PONE-D-25-65253Unraveling the significance of DCN in endometriosis development through single cell sequencing and experimental approachesPLOS One

Dear Dr. Wang,

Thank you for submitting your manuscript to PLOS ONE. After careful consideration, we feel that it has merit but does not fully meet PLOS ONE’s publication criteria as it currently stands. Therefore, we invite you to submit a revised version of the manuscript that addresses the points raised by the revisors during the review process.

We look forward to receiving your revised manuscript.

Kind regards,

Stefania Crispi

Academic Editor

PLOS One

Journal Requirements:

“Beijing Jishuitan Hospital Youcai Plan (JSTYC202402) .”

4. Please be informed that funding information should not appear in the Acknowledgments section or other areas of your manuscript. We will only publish funding information present in the Funding Statement section of the online submission form. Please remove any funding-related text from the manuscript.

5. We notice that your supplementary figures are uploaded with the file type 'Figure'. Please amend the file type to 'Supporting Information'. Please ensure that each Supporting Information file has a legend listed in the manuscript after the references list.

Reviewers' comments:

Reviewer's Responses to Questions

**Comments to the Author**

1. Is the manuscript technically sound, and do the data support the conclusions?

Reviewer #1: Yes

Reviewer #2: Yes

2. Has the statistical analysis been performed appropriately and rigorously? 

Reviewer #1: Yes

Reviewer #2: I Don't Know

3. Have the authors made all data underlying the findings in their manuscript fully available?

Reviewer #1: Yes

Reviewer #2: Yes

4. Is the manuscript presented in an intelligible fashion and written in standard English?

Reviewer #1: Yes

Reviewer #2: Yes

5. Review Comments to the Author

Reviewer #1: The manuscript by Chen et al. investigates the role of Decorin (DCN) in endometriosis by integrating single-cell RNA sequencing, bulk transcriptomic deconvolution, and in vitro validation. The study addresses a relevant and timely topic, given the unmet need for reliable biomarkers and mechanistic insights into endometriosis pathogenesis. The combination of scRNA-seq–based stromal cell subtyping with experimental validation represents a notable strength.

Overall, the manuscript is interesting, methodologically solid, and potentially suitable for publication, but several conceptual, methodological, and presentation issues need to be addressed before acceptance.

Major comments

1. Conceptual clarity on the role of DCN (biomarker vs functional driver)

While the study convincingly demonstrates differential expression and diagnostic potential of DCN, the manuscript repeatedly refers to DCN as a “promoter of endometriosis progression”

However:

o The functional experiments are limited to expression validation (RT-qPCR).

o No loss-of-function or gain-of-function assays (e.g., DCN knockdown, overexpression, rescue experiments) are presented.

o No direct mechanistic assays (e.g., migration, invasion, fibrosis markers, TGF-β signaling readouts) are shown.

2. Stromal subcluster annotation requires stronger validation

The identification of 10 stromal subtypes is a central result.

However:

o Functional annotation is based mostly on enrichment analysis rather than well-established stromal markers. Provide a clear table of top 5–10 marker genes per subtype to support annotations.

o Names such as Stemness-Remodeling Stromal Cells and Metabolic-Endocrine Dual-Function Stromal Cells are introduced without literature precedent. Clarify that these names represent functional states rather than definitive identities, or move them to supplementary material.

o The manuscript does not sufficiently compare stromal subtypes with previously described endometrial stromal populations. Explicitly compare the identified subtypes with published human endometrium or endometriosis scRNA-seq datasets to strengthen biological validity.

3. Inconsistencies in immune terminology

o The manuscript refers to immune microenvironment analysis, but deconvolution focuses mainly on stromal subtypes, not classical immune cells.

o Terms such as “immune cell proportion” are used inconsistently (Figures 4C–D).

Clarify terminology throughout. Use “cellular composition” or “stromal composition” instead of immune cells where appropriate.

Justify the use of CIBERSORTx for stromal deconvolution, as it is typically applied to immune populations.

4. Diagnostic value of DCN: limited clinical applicability

o ROC analyses show good AUC (>0.8), but sample sizes are small, particularly for controls (e.g., GSE135485).

o No independent clinical validation cohort is included.

o Tissue-based expression may limit clinical translation.

Avoid overstatement of DCN as a diagnostic biomarker.

Discuss potential non-invasive applications, e.g., detection in serum, peritoneal fluid, or menstrual effluent.

5. Overinterpretation of drug prediction results

o Seven DCN-targeting drugs are identified, but no mechanistic evidence or prior validation is provided.

o Some compounds (e.g., sirolimus, interferon) have broad systemic effects.

State clearly that this analysis is hypothesis-generating only.

Avoid implying therapeutic applicability without experimental validation.

Reviewer #2: This study presents interesting data identifying stromal cells as a key population in endometriosis, identifying 10 subtypes via secondary clustering. A derived signature matrix effectively predicted cellular composition in bulk RNA datasets, revealing significant proportional differences in specific subtypes (AMSC, FESC, SRSC, and dStromal) between disease and control groups. Additionally, DCN was identified as a key differentially expressed gene—validated by RT-qPCR—with high diagnostic potential

Minor revision

• The manuscript overstated to investigate the role and mechanism of DCN in endometriosis development. However, the data presented is strictly limited to characterizing the expression profile, clinical relevance (ROC curves) and identifying potential signaling pathways via GESA. There are no functional experiments (e.g., Knockdown/Overexpression of DCN) to verify if DCN modulation actually alters cell migration or fibrosis. The lack of mechanistic investigation of DCN role is only clarified in the limitations section. Please revise the Abstract and text to accurately reflect the scope of the work. The authors should explicitly state that this is a bioinformatic characterization and correlation study, and tone down assertions regarding DCN's definitive role in pathogenesis until functional validation is provided.

• In the discussion section, you reference “Aydin GA et al.” findings regarding the link between DCN in VEGF in endometriosis-associated angiogenesis. However, you don’t explain the nature of this relationship or how it connects to your current findings. Please clarify specifically how DCN interacts with VEGF in this context.

• In the summary paragraph, you mention specific downstream targets of DCN such as CSF1R, MMP2, and MMP9 that weren’t previously discussed in the body of the manuscript. It is confusing to present these targets as conclusions without providing evidence in the preceding text. Please revise the discussion section to ensure all points raised in the summary and conclusion are fully established in the main text.

• The figures are pixelated and difficult to interpret. Please replace them with higher-quality images.

6. PLOS authors have the option to publish the peer review history of their article (what does this mean?). If published, this will include your full peer review and any attached files.

Reviewer #1: No

Reviewer #2: **Yes:** Hebatallah Hassan

You may also use PLOS’s free figure tool, NAAS, to help you prepare publication quality figures: https://journals.plos.org/plosone/s/figures#loc-tools-for-figure-preparation

---

## [Author Response · Author response to Decision Letter 1]

6 Apr 2026

Dear Reviewers,

Thank you for your thoughtful suggestions and insights, which have benefited from the manuscript. I am looking forward to working with you to move this manuscript closer to publication in “PLOS One”.

The manuscript has been rechecked and the necessary changes have been made in accordance with your suggestions. The responses to all comments have been prepared and attached below. We have tried our best to solve the problems you proposed, and we hope that the revised manuscript is now suitable for publication in the journal “PLOS One”. If you have any questions remained about this paper, please feel free to contact us.

---

## [Decision Letter · Decision Letter 1]

30 Apr 2026

Unraveling the significance of decorin in endometriosis development through single cell sequencing and experimental approaches

PONE-D-25-65253R1

Dear Dr. Wang,

We’re pleased to inform you that your manuscript has been judged scientifically suitable for publication and will be formally accepted for publication once it meets all outstanding technical requirements.

Kind regards,

Stefania Crispi

Academic Editor

PLOS One

Additional Editor Comments (optional):

Reviewers' comments:

Reviewer's Responses to Questions

**Comments to the Author**

1. If the authors have adequately addressed your comments raised in a previous round of review and you feel that this manuscript is now acceptable for publication, you may indicate that here to bypass the “Comments to the Author” section, enter your conflict of interest statement in the “Confidential to Editor” section, and submit your "Accept" recommendation.

Reviewer #1: All comments have been addressed

Reviewer #2: All comments have been addressed

2. Is the manuscript technically sound, and do the data support the conclusions?

Reviewer #1: Yes

Reviewer #2: (No Response)

3. Has the statistical analysis been performed appropriately and rigorously? 

Reviewer #1: Yes

Reviewer #2: (No Response)

4. Have the authors made all data underlying the findings in their manuscript fully available?

Reviewer #1: Yes

Reviewer #2: (No Response)

5. Is the manuscript presented in an intelligible fashion and written in standard English?

Reviewer #1: Yes

Reviewer #2: (No Response)

6. Review Comments to the Author

Reviewer #1: I appreciate the authors’ thorough revisions and detailed responses. The concerns raised have been adequately addressed, and the manuscript has been significantly improved. I believe it is now suitable for publication.

Reviewer #2: The authors have addressed all my previous concerns, and the manuscript has significantly improved. I recommend the paper for publication in its current form, but only after fixing one minor error at lines (343-344), Page (17) in the discussion section. Since no functional experiments were conducted in this paper, please update the text to state that these will be done in future studies.

7. PLOS authors have the option to publish the peer review history of their article (what does this mean?). If published, this will include your full peer review and any attached files.

Reviewer #1: No

Reviewer #2: No

---

## [Editor Report · Acceptance letter]

PONE-D-25-65253R1

PLOS One

Dear Dr. Wang,

I'm pleased to inform you that your manuscript has been deemed suitable for publication in PLOS One. Congratulations! Your manuscript is now being handed over to our production team.

Kind regards,

on behalf of

Dr. Stefania Crispi

Academic Editor

PLOS One